

# Middle East Respiratory Syndrome Coronavirus and the One Health concept

Maged Gomaa Hemida

Department of Microbiology and Parasitology, College of Veterinary Medicine, King Faisal University, Al-Hufuf, Al-Hasa, Saudi Arabia
Department of Virology, faculty of veterinary medicine, Kafrelsheikh University, Egypt, Kafrelsheikh University, Kafrelsheikh, Kafrelsheikh, Egypt

## ABSTRACT

Middle East Respiratory Syndrome Coronavirus (MERS-CoV) is one of the major threats to the healthcare systems in some countries, especially in the Arabian Peninsula. MERS-CoV is considered an ideal example of the One Health concept. This is due to the animals, especially dromedary camels, play important roles in the transmission and sustainability of the virus, and the virus can be transmitted through aerosols of infected patients into the environment. However, there is some debate regarding the origin of MERS-CoV either from bats or other unknown reservoirs. The dromedary camel is the only identified animal reservoir to date. These animals play important roles in sustaining the virus in certain communities and may act as an amplifier of the virus by secreting it in their body fluids, especially in nasal and rectal discharges. MERS-CoV has been detected in the nasal and rectal secretions of infected camels, and MERS-CoV of this origin has full capacity to infect human airway epithelium in both in vitro and in vivo models. Other evidence confirms the direct transmission of MERS-CoV from camels to humans, though the role of camel meat and milk products has yet to be well studied. Human-to-human transmission is well documented through contact with an active infected patient or some silently infected persons. Furthermore, there are some significant risk factors of individuals in close contact with a positive MERS-CoV patient, including sleeping in the same patient room, removing patient waste (urine, stool, and sputum), and touching respiratory secretions from the index case. Outbreaks within family clusters have been reported, whereby some blood relative patients were infected through their wives in the same house were not infected. Some predisposing genetic factors favor MERS-CoV infection in some patients, which is worth investigating in the near future. The presence of other comorbidities may be another factor. Overall, there are many unknown/confirmed aspects of the virus/human/animal network. Here, the most recent advances in this context are discussed, and the possible reasons behind the emergence and sustainability of MERS-CoV in certain regions are presented. Identification of the exact mechanism of transmission of MERS-CoV from camels to humans and searching for new reservoir/s are of high priority. This will reduce the shedding of the virus into the environment, and thus the risk of human infection can be mitigated.

Corresponding author
Maged Gomaa Hemida,
mhemida@kfu.edu.sa,
gomaa55@gmail.com

## INTRODUCTION

The main reason behind developing this article is to summarize the current understanding about MERS-CoV in the context of the One Health concept. In this article, I highlight the known information about the MERS-CoV infection and its pathogenesis in humans, the patterns of MERS-CoV in dromedary camels, the potential roles of other animals in the transmission cycle of MERS-CoV, and the interaction of MERS-CoV/humans/animals. I elaborate on how some strategies can be used to stop or reduce the frequencies of MERS-CoV outbreaks based on the One Health concept, identified some gaps in the literature, and drew conclusions. The One Health concept established to ensure the good health and well beings of the human, animal and the environment. Human health is mainly affected by environment health as well as animal health (*Lerner & Berg, 2015*).

### Survey methodology

For this review article, I conducted a literature search of the most up-to-date published articles on MERS-CoV in the past 7 years. First, I focused the introduction section on the historical background of coronaviruses and the One Health concept. Then, I highlighted the most up-to-date literature from PubMed central, Google Scholar and ResearchGate on the interaction of MERS-CoV/humans/animals. I identified some important gaps in the research dealing with MERS-CoV/human/environment in the context of the One Health concept. I also summarized the current acceptable theories on the emergence and evolution of MERS-CoV. Finally, I highlighted progress to date in the control of MERS-CoV. Historically, MERS-CoV was first identified in Saudi Arabia in a patient suffered from severe pneumonia and shortening of breath. The virus was called the novel coronavirus at that time (*Zaki et al., 2012*). Another retrospective study conducted in Jordan early 2012 revealed the detection of this novel coronavirus in 11 patients. Eight out of them were from the health care workers (*Hijawi et al., 2013*).

## CORONAVIRUSES: THE PAST, PRESENT, AND FUTURE

Coronaviruses are a large group of viruses causing many health problems (respiratory, enteric, and nervous syndromes) in various species of animals and humans. Six human coronaviruses that have been identified to date (HCoV-229E, HCoV-OC43, HCoV-NL-63, HCoV-HUK-1, SARS-CoV, and MERS-CoV). Two out of them emerged in the past 15 years (*Lau & Chan, 2015*), namely, the severe acute respiratory syndrome coronavirus (SARS-CoV) and the Middle East Respiratory Syndrome Coronavirus (MERS-CoV). SARS-CoV emerged in 2003 in China and spread to many countries throughout the world (*Peiris et al., 2003*). Approximately 8,000 people were infected, and 10% of them died (*Aronin & Sadigh, 2004*). Only 9 years later, MERS-CoV emerged in Saudi Arabia (*Zaki et al., 2012*). This is a relatively short period for the emergence of a new coronavirus. One of the main reasons behind the rapid emergence of new coronaviruses is the poor proofreading capability of their RNA polymerases (*Hofer, 2013*). This is in addition to the possibility of the recombination of different coronaviruses (*Makino et al., 1986*), and it will not be surprising if new coronaviruses emerge in the near future. MERS-CoV continues to pose

great challenges to the healthcare system of some countries in the Middle East and Arabian Peninsula. Since its discovery late in 2012 (*Zaki et al., 2012*), there are ongoing reports to the World Health Organization (WHO) from some countries in the Middle East, especially the Arabian Peninsula, with spread to other countries around the globe. According to the latest WHO statistics, there have been a total of 2,428 laboratory-confirmed cases of MERS-CoV infection including at least 838 deaths (reported case fatality rate of 35.0%) (*WHO, 2018*). The continuous ongoing reports on MERS-CoV suggesting the presence of some factors favor its sustainability in certain regions. There are many uncertain aspects of the virus evolution, pathogenesis, and transmission cycle. Unfortunately, recently, there was some decline in the rate of research on the virus from different aspects (*Hemida et al., 2017b*). This hampered the production of new data about the MERS-CoV from different aspects. Below, I summarize the current understanding of the virus in the context of the One Health concept.

## CORONAVIRUSES AND THE ONE HEALTH CONCEPT

The One Health concept is an interesting concept outlining the close interaction among humans, animals and the environment (*Destoumieux-Garzon et al., 2018*). Currently, there are two coronaviruses candidates representing the One Health concept, SARS-CoV and MERS-CoV. Animals play important roles in the transmission cycle of both viruses (*Alshukairi et al., 2018*; *Wang et al., 2005*). Both viruses were proven to be of zoonotic origin (*Gao et al., 2016*). The palm civet cat played important role in the transmission cycle of SARS-CoV (*Wang et al., 2005*). Some patients proved to visit one restaurant serving the civet cats as a meal (*Wang et al., 2005*). Culling of the civet cats resulted in marked decline in the reported SARS-CoV cases and now become extinct. Many studies made a direct link between the exposure to camel and its meat and milk products and MERS-CoV human cases (*Reusken et al., 2014*). Several studies reported the presence of MERS-CoV specific antibodies in sera of human came in close contact with camels (*Reusken et al., 2014*; *Reusken et al., 2016*). Meanwhile, MERS-CoV was isolated from air pf positive dromedary camel herds in Saudi Arabia (*Azhar et al., 2014*).

## MERS-COV IN HUMANS

MERS-CoV may infect a wide group of people ranging from very young ages, even infants less than one year of age, to 109 years of age (*CDC, 2016*). However, children are less likely to be infected with MERS-CoV when compared to adults and, if infected, they tend to have asymptomatic or mild disease (*Arwady et al., 2016*). The reason for this is still not entirely clear and requires further study.

The case fatality rate is always very high in case of the immunocompromised infected patients especially those who are suffering from chronic diseases such as cancer, diabetes, blood pressure, kidney problems, etc. (*Arwady et al., 2016*).

Human-to-human transmission is reported in many cases. MERS-CoV replicates efficiently in various *in vitro* and *ex vivo* models (*Chan et al., 2014*). Moreover, many family clusters and hospital outbreaks were reported in the past 5 years (*Arwady et al.,*

2016; *Drosten et al., 2014*; *Memish et al., 2013*). This confirms the potential spread of MERS-CoV among those who are in close contact in the population (*Mollers et al., 2015*). The most at-risk groups are healthcare workers including nurses, medical doctors and other hospital staff and the elderly with underlying chronic diseases (*Arabi et al., 2014*).

The prevalence rate of MERS-CoV in primary cases among males is relatively higher than that of females (*Darling et al., 2017*), which may be because exposure to infected dromedary camels is much higher in males than in females.

MERS-CoV infection triggers some unique interferons and cytokine gene expression profiles. The virus seems to be a poor interferon inducer (*Chan et al., 2014*). This suggests the potential immune evasion strategies triggered by the virus to hijack the host immune system and may be responsible for the high fatality rate, at least in part. Viral spreading among people seems to not yet be very efficient. Those in close contact are among the at-risk groups for infection (*Drosten et al., 2014*), as observed in many hospital outbreaks as well as family clusters (*Alfaraj et al., 2018*; *Choi et al., 2017*; *Xiao et al., 2018*). This suggests that transmission of the virus among people requires exposure to a high viral load, which will sometimes produce active infection in people who are in close contact. Several MERS-CoV family clusters have been reported (*Drosten et al., 2014*). Interestingly, MERS-CoV is reported in the dromedary camels in many African countries (Egypt, Nigeria, Tunisia, and Ethiopia), but no primary human cases have been reported in these countries to date (*Ali et al., 2017*; *Roess et al., 2016*; *Van Doremalen et al., 2017*), which may be related to some variation in the circulating Asian and African strains of MERS-CoV.

Some important deletions in the MERS-CoV currently circulating in dromedary camels from Africa were recently reported (*Chu et al., 2018*). These deletions may explain at least in part the reason behind the variations in the pathogenesis among the Asian and African strains of MERS-CoV. Another potential reason behind the absence of human cases in the African countries is the diverse cultural habits among people in Africa and the Arabian Peninsula (*FAO, 2016*). People in Arabian Peninsula get in more close touch with camels during the camel show, sports, trade than in Africa. This make the human risk of exposure much higher in the AP than Africa. MERS-CoV infection varies from severe respiratory illness accompanied by a high fever and respiratory distress to mild asymptomatic cases. Patients are usually admitted to the intensive care unit (ICU) and provided with a source of oxygen. Most cases result in pneumonia, which is fatal in almost 40% of the affected patients (*Hong et al., 2017*; *Rubio et al., 2018*). Some patients may develop renal failure [13]. Several MERS-CoV travel-associated infections were in many cases associated with the Middle East (*Bayrakdar et al., 2015*; *Rubio et al., 2018*). Among these reported was the Korean outbreak in early 2015 (*Choi et al., 2017*; *Kim, Andrew & Jung, 2017*; *Xiao et al., 2018*). One Korean citizen visited some countries in the Middle East and then returned home ill. This person visited several healthcare facilities in Korea. This resulted in the largest MERS-CoV human outbreak outside the Arabian Peninsula (AP) (*Xiao et al., 2018*). This outbreak confirmed the human-to-human transmission. During this outbreak, MERS-CoV was isolated from air samples from the hallways of the healthcare facilities close to the hospitalized patients (*Xiao et al., 2018*). This at least explains in part the rapid development of MERS-CoV hospital outbreaks.

## MERS-COV IN ANIMALS

Since the discovery of MERS-CoV late in 2012 (*Zaki et al., 2012*), many research groups have searched for its potential animal reservoir/s. Dromedary camels are the only currently proven reservoir for MERS-CoV (*Hemida et al., 2014*; *Hemida et al., 2017b*; *Reusken et al., 2014*; *Reusken et al., 2016*). Interestingly, others were able to trace the virus back 30 years ago in dromedary camel specimens in retrospective studies (*Corman et al., 2014*; *Hemida et al., 2014*; *Reusken et al., 2014*). All these data suggest that the virus has been circulating for decades without being recognized. Although the actual and typical clinical features of the MERS-CoV natural infection in dromedary camels is not well documented to date, very few studies reported these patterns under experimental infection approaches (*Adney et al., 2014*). Based on these findings, camels do not show any pathognomonic signs despite a subtle fever and mild nasal discharge for up to 6 days post-infection (dpi) (*Hemida et al., 2014*). Meanwhile, shedding of the infectious virus was reported in the experimentally infected camels started at 2 dpi up to the 7th dpi (*Adney et al., 2014*). Interestingly, viral RNA was still detected at 35 dpi (*Adney et al., 2014*), though it is not known whether the viral RNAs may act as potential sources of infection similar to some other positive-sense RNA viruses. No viral shedding in the oral secretions, rectal swabs, urine, or sera of these animals has been reported (*Adney et al., 2014*), in contrast to the detection of the virus in the fecal specimens and swabs under natural viral field infection (*Hemida et al., 2014*). These finding suggesting differential patterns of MERS-CoV infection between natural and experimental approaches. Further studies are required to understand the natural MERS-CoV infection in dromedary camels, which may be achieved by conducting long-term longitudinal studies as well as careful monitoring of the virus infection in large populations of camels. On necropsy examination of MERS-CoV, experimentally infected dromedary camels revealed only mild-to-moderate inflammatory reactions in the upper respiratory tract (*Khalafalla et al., 2015*). Detection of the viral antigens in the tissue sections of the turbinate bone and the upper respiratory tract was reported (*Adney et al., 2014*). Interestingly, seroconversion of the inoculated animals was reported to begin at 14 dpi (*Hemida et al., 2014*), indicating that MERS-CoV induces a robust humoral immune response after infection. More recently, one longitudinal study reported the possibility of MERS-CoV infection in seropositive animals. This raises concern about the role of the antibodies in the protection of the MERS-CoV infection (*Hemida et al., 2017a*). It seems that all the members of the family *Camelidae* (dromedary, alpaca, and llamas) are susceptible to MERS-CoV infection (*Corman et al., 2014*; *Vergara-Alert et al., 2017*). *David et al. (2018)* reported the presence of antibodies against MERS-CoV in some alpacas and llamas in Israel but only used commercial ELISA kits, and they did not address the possibility of cross-reactivity with other coronaviruses especially BCoV. It had been previously showed that there is clear cross-reactivity between MERS-CoV and BCoV in dromedary camels (*David et al., 2018*). Interestingly, one study showed an absence of any detectable antibodies of MERS-CoV in the sera of Bactrian camels (*Chan et al., 2015*), though this is the only study to report this finding concerning the seronegativity of Bactrian camels to MERS-CoV. It is not well known whether the absence of detectable MERS-CoV antibodies in the sera of Bactrians camels is due to the

geographical location of the tested animals in Mongolia, far from the Middle East and Africa. This may be supported by similar findings in dromedary camels in Australia and the Canary Islands (*Crameri et al., 2015*). Another possibility is that this might be due to some genetic factors, which contribute to the resistance of Bactrians to MERS-CoV infections; this point is worthy of further investigation. Experimental MERS-CoV infection in both alpacas and llamas showed a similar pattern to that of dromedary camels (*Crameri et al., 2016*; *Vergara-Alert et al., 2017*), which suggested that both animals might act as a model animal for the study of MERS-CoV *in vivo*. However, the experimental infection of pigs with MERS-CoV did not reveal as much infection as that reported in alpacas and llamas (*Vergara-Alert et al., 2017*). Active MERS-CoV particles were neither retrieved from the experimentally infected animals nor from the close contact non-infected animals during the duration of this study (*Vergara-Alert et al., 2017*). This result suggested that pigs might not play an active role in the transmission of MERS-CoV. Although bats are considered the main reservoir for many coronaviruses, their roles in the MERS-CoV still need further clarifications. One study reported the presence of MERS-CoV in one specimen collected from bats in Saudi Arabia (*Memish et al., 2013*). The genome sequence of this particular virus showed almost 100% identity to a MERS-CoV index case (*Memish et al., 2013*). More recently, Jamaican fruit bats were found to be permissible for MERS-CoV infection (*Munster et al., 2016*). However, MERS-CoV-infected bats did not show any apparent clinical signs; however, viral shedding was reported in the swabs from bats up to 9 dpi. The clinical profiles and viral shedding curve during the course of the MERS-CoV infection in these bats look similar to that of dromedary camels (*Munster et al., 2016*), yet the amount of infectious viral shedding in bats is less compared to that in dromedary camels. This species of bat is not the most relevant for MERS-CoV infection, but this study offers some insights into the molecular pathogenesis of MERS-CoV in bats. Interestingly, another study revealed the expression of MERS-CoV receptors (dipeptidyl peptidase-4, DPP4) in the respiratory and digestive tracts of some insectivorous bats (*Widagdo et al., 2017*). An interesting study tested the potential roles of other species in the transmission of MERS-CoV such as cattle, sheep, goats, donkeys, buffaloes, mules, and horses from Egypt, Tunisia and Senegal (*Kandeil et al., 2019*). This study revealed the presence of neutralizing antibodies in the sera of some sheep and goats. Meanwhile, viral RNA was detected in swabs collected from some sheep, goats and donkeys from these countries (*Kandeil et al., 2019*). Several attempts were made to identify an appropriate experimental animal model for MERS-CoV. The Syrian hamster was non-permissive to MERS-CoV infection (*De Wit et al., 2013*). Experimental infection of this animal neither develops any clinical signs or pathology nor produces any cytokines after infection (*De Wit et al., 2013*). This was in contrast to New Zealand white rabbits, which showed active infection after inoculation with the MERS-CoV (*Monchatre-Leroy et al., 2017*). Furthermore, both rhesus macaques and common marmosets supported MERS-CoV infection (*Yu et al., 2017*). Additionally, both the transgenic and the transduced mice expressing the dipeptidyl peptidase 4 human receptors worked as a model for MERS-CoV studies (*Zhao et al., 2015*). Interestingly, a new study reported the seropositivity of some sheep and goat to MERS-CoV from Tunisia, Senegal and Egypt (*Kandeil et al., 2019*). Same study reported the detection of the viral

RNAs in samples from cow, sheep, goat and donkeys from Egypt (*Kandeil et al., 2019*). This highlights the importance of continuous surveillance and searching for new reservoir/s for the MERS-CoV transmission cycle.

## MERS/HUMAN/ANIMAL INTERACTION

It is now well accepted that human exposure to MERS-CoV-infected dromedary camels is a predisposing factor to human infection, particularly in immunocompromised people (*Zumla et al., 2015*). Based on the latest WHO reports, the prognosis of MERS-CoV infection is poor for elderly patients who have chronic diseases such as cancer, diabetes, kidney failure, etc. (*Arabi et al., 2014*). Transmission of MERS-CoV from dromedary camels to humans has been proven indirectly in some previous reports (*Azhar et al., 2014*). One study showed strong evidence of direct transmission of MERS-CoV from an infected camel to its owner, which was confirmed by comparing the viral genome sequencing of the virus isolated from the infected dromedary camel to that isolated from its owner. Both viruses were almost a 100% match (*Azhar et al., 2014*). Meanwhile, this study reported the detection of MERS-CoV nucleic acid in air samples from the indicated dromedary camel barn during the active course of the viral infection (*Azhar et al., 2014*).

There is a debate about the role of the dromedary camel's milk and meat products and by-products in the transmission of MERS-CoV. Experimental introduction of MERS-CoV to raw milk revealed little difference between the virus stock in milk and that kept in DMEM (*Van Doremalen et al., 2014*). Due to their culture, some people in the Middle East would drink raw camel milk in efforts to seek treatment for some diseases such as diabetes. The authors acknowledge that MERS-CoV was introduced into the dromedary camel milk at a high dose and that the viral RNA was detected in a limited concentration in the camel's milk (*Van Doremalen et al., 2017*). Thus, drinking the raw camel milk poses a great risk to the people consuming this milk without any heat treatment or pasteurization (van Doremalen et al. 2014; (*Zhou et al., 2017*). One study connected the infection of some people to the drinking of the milk from one infected camel (*Memish et al., 2015*). However, another study was conducted in Qatar to assess the possibility of acquiring the infection from contaminated teats and udders of infected she-camels during the process of milking (*Reusken et al., 2014*), though no active MERS-CoV shedding in milk has yet to be reported. Further studies are encouraged to come to a conclusion about the potential role of raw camel milk in the transmission of MERS-CoV. Nonetheless, the role of camel meat in the transmission of MERS-CoV has not been studied carefully to date. Thus, special attention should be paid to the efficient cooking of camel meat and its products as well as thorough boiling of camel milk. It is suggested that people not drink raw camel milk to avoid any risk of infection not only with MERS-CoV but also with other pathogens such as Brucellosis (*Garcell et al., 2016*). Some studies reported that MERS-CoV is one of the occupational zoonotic viral diseases, as was claimed based on some studies investigating the seroconversion of some at-risk group of people to MERS-CoV. This study reported the presence of specific MERS-CoV antibodies in approximately 3% of the workers in some slaughterhouses in Qatar (*Farag et al., 2015*). On the other hand, some studies reported
the absence of any detectable antibodies in the sera of some herdsmen, veterinarians, and slaughterhouses in Saudi Arabia (*Hemida et al., 2015*). One possible explanation for the variations between the two studies is the difference in the sensitivity of the techniques used. Those two studies used two different techniques to report the presence/absence of the MERS-CoV antibodies in the sera of at-risk people (*Farag et al., 2015*; *Hemida et al., 2015*). Regardless, further investigation on a large-scale basis is required to solidify this conclusion about MERS-CoV.

## GAPS ON THE MERS-COV-RELATED RESEARCH

There is more to be known about the molecular biology of MERS-CoV. Identification of the DPP-4 as viral receptors does not rule out the presence of other co-receptors or transcription/translation factors that favor the virus infection in a certain host. There are many immune evasion strategies triggered by MERS-CoV to hijack the host immune responses, and the mechanisms of such strategies have not been well studied. Moreover, there are many unknown aspects especially in the context of the MERS-CoV/human/animal interaction. Meanwhile, some studies were conducted on a small scale or with low numbers of animals/specimens and reported some important conclusion. These studies need further confirmation, and refinement of some of these observations is urgently needed in the near future. Here, we highlight some gaps in the research regarding the evolution and transmission of MERS-CoV. Presumably, there might be an unidentified reservoir in the transmission cycle of MERS-CoV. Although respiratory infection still is the main route of MERS-CoV infection, the exact mechanism of transmission of MERS-CoV from dromedary camels to humans is still not well understood. The possibility of another reservoir in the transmission cycle of MERS-CoV has not been ruled out. Thus, there might be a missing link in the chain of the human/camel interaction. Meanwhile, the exact modes of transmission of MERS-CoV from dromedary camels to humans have not been well clarified, and the typical pattern of the natural MERS-CoV infection in dromedary camels has not been well studied. Additionally, the potential role of most camel secretions and excretions has not been fully understood. The seroprevalence of MERS-CoV was reported in the dromedary camels from different countries in Africa and Asia (*Ali et al., 2017*; *Hemida et al., 2014*), though feral camels in Australia and the Canary islands were found to be seronegative (*Crameri et al., 2015*). The reason behind this phenomenon may be due to these regions being isolated lands and away from the MENA region as described above; it may also be due to the absence of an active camel movement between the Middle East and Africa and these regions of the world. Very few studies reported the cross-reactivity between MERS-CoV and other coronaviruses such as the bovine coronavirus (BCoV) that might infect dromedary camels. It is unknown if this might be the reason behind the high seroprevalence of MERS-CoV among the dromedary camel population. This may be due to the high frequency of exposure to the MERS-CoV infection during the camel's life, the cross-reactivity of other coronaviruses, or an unknown mechanism related to the dromedary camel's immune system. These considerations require further studies. There is ongoing demand for the development of novel diagnostic assays for coronaviruses, and

special interest should be paid to those techniques that enable the simultaneous detection of the viral nucleic acids and those that can simultaneously distinguish between the antibodies for several coronaviruses. Furthermore, it is not well understood why only the Bactrian camels among the family Camelidae did not seroconvert to MERS-CoV infection (*Chan et al., 2015*). The genetic susceptibility of certain human populations, especially blood-related people, is not clear in the context of MERS-CoV infection. There are several levels of human exposure to dromedary camels, such as camel attendants, workers in camel abattoirs, veterinarians inspecting their carcasses, and camel owners. Those groups of people are in close contact with camels for various amounts of time and are considered to be a high-risk group of people due to the long time they spend in close contact with the dromedary camels. Meanwhile, there is an urgent need to develop a risk scoring system for human exposure to the dromedary camels.

## CURRENT THEORIES IN THE MERS-COV/HUMAN/ANIMALS INTERACTION

It is believed that there is some unidentified reservoirs in the context of MERS-CoV transmission presenting the virus to the community. This virus is able to infect dromedary camels, which act as an amplifier host for the virus, favoring the circulation of the virus in some camel herds. The virus has the ability to circulate among the animals in the same herd and the camel herds in close proximity to them (Fig. 1). MERS-CoV in camels has the full potential to infect the human especially immunocompromised persons. Once the virus infects a human, there is always a possibility of infecting other people, especially closely-related individuals (Fig. 1), including household relatives and workers plus healthcare workers such as doctors and nurses. Infection depends on the level of exposure to the infected person, and MERS-CoV infection in humans ranges from very severe cases of pneumonia to death. Currently available data indicate that severely infected individuals can shed the infectious virus into the environment (*Kim et al., 2016*), though there are few data regarding the capacity of mildly infected individuals to transmit the virus. Asymptomatic individuals, however, are unlikely to transmit the virus (*Moon & Son, 2017*).

## POTENTIAL REASONS FOR THE EMERGENCE AND THE SPREAD OF MERS-COV

There are many factors behind the emergence, sustainability and spread of MERS-CoV. The presence of an unidentified MERS-CoV reservoir in the transmission cycle is still considered, and this unknown reservoir may contribute substantially to the suitability of the virus in certain regions. Dromedary camels remain the amplifier of the virus; the close contact of these animals to the human population in certain regions of Africa and Asia may pose a great risk for human infection and indirectly contribute to the spread of the virus. Additionally, public animal markets, especially for dromedary camels, may act as an amplifier of the virus. This poses a great risk to the surrounding community. Interesting study addressed the mapping of MERS-CoV cases in association with the environmental conditions and camel exposure (*Reeves, Samy & Peterson, 2015*). This study revealed that

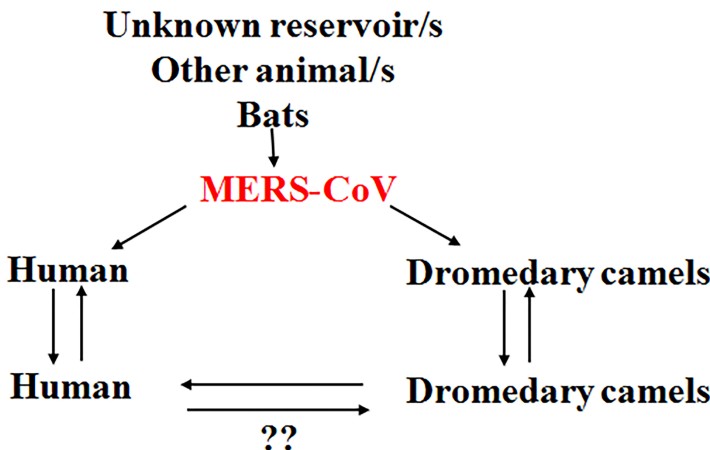

**Figure 1 Current theories regarding the MERS-CoV/human/animal interaction.** There might be an unknown reservoir in the transmission cycle of MERS-CoV. Bats play a role in the context of MERS-CoV transmission. The virus is transmitted to dromedary camels through an unknown mechanism. The dromedary camels act as amplifying hosts for the virus. MERS-CoV is transmitted from dromedary camels to humans through the respiratory aerosols and some other unknown mechanisms. The virus is then transmitted among the human population through respiratory routes. The human-to-human transmission has been confirmed. The human-to-camel transmission still needs further clarification. Question marks indicate the non-confirmed phenomenon.

camel exposure is a key predisposing factor for some of MERS-CoV human cases (*Reeves, Samy & Peterson, 2015*). The lack of active surveillance programs for respiratory viruses, especially coronaviruses, may result in many subclinical or mild cases of MERS-CoV being missed in a certain population. These patients may shed the virus in their secretions and may act as a source of infection to other persons in close contact with them. Although many MERS-CoV vaccine and drug candidates are being researched, none are available yet. All these factors may favor the sustainability of MERS-CoV in certain regions.

## CURRENT PROGRESS ON THE CONTROL OF MERS-COV

Interestingly, the case fatality rate of MERS-CoV among the affected population dropped from almost 50% in 2012 to 34% early 2019 (*WHO, 2018*), and we may relate this progress in the control of MERS-CoV over the past 7 years to many factors. First, identification of the main reservoir of the virus, namely, the dromedary camel (*Hemida et al., 2014*). Second, continuous molecular and serological surveillance of MERS-CoV among the dromedary camel population in the Arabian Peninsula and Africa (*Corman et al., 2014*; *Farag et al., 2015*; *Hemida et al., 2017a*; *Hemida et al., 2017b*; *Khalafalla et al., 2015*; *Reusken et al., 2014*). Currently, testing the population of camels in regional camel markets is associated with shutting down of the market in case of positive shedding of MERS-CoV by the animals. I believe this will substantially minimize the risk of community-acquired infections through these positive populations. Third, vaccination of dromedary camels, especially animals under two years of age, will have a great impact on the reduction of the viral shedding from these animals to the surrounding community. This will also have a great positive impact on

the reduction of the number of reported human infections. Fourth, there has been progress in the current understanding of viral tropism, pathogenesis, and mode of transmission in the past five years (*Chan et al., 2014*; *Widagdo et al., 2017*). Fifth, new strategies have been adopted to reduce the spread of infection in health care units (*Rajakaruna et al., 2017*). Sixth, some therapeutic and control approaches for MERS-CoV such as cyclosporine, ribavirin and interferon show promising trends for the treatment of MERS-CoV-infected patients (*Al-Tawfiq et al., 2014*; *De Wilde et al., 2013*). Meanwhile, good progress has been made in screening large numbers of drugs/therapies for the treatment of MERS-CoV (*Han et al., 2018*; *He et al., 2019*; *Niu et al., 2018*; *Totura & Bavari, 2019*). This may lead to the development of some effective novel drugs against MERS-CoV infection in the near future.

## ONE HEALTH-BASED INTERVENTIONS TO STOP MERS-COV OUTBREAKS

To stop MERS-CoV outbreaks, there are several strategies to be adopted in the context of the One Health concept. Some strategies are related to the animal, while others are related to human health. The main objective is to minimize or stop the viral shedding from dromedary camels to the environment (Fig. 2). This may be achieved in many ways including regular monitoring of the population of dromedary camels. Active animal shedders need to be identified, and quarantine measures should applied until they stop shedding the virus. Vaccination of young dromedary camel calves should occur during their first 6 months of life, which will minimize the chances of these animals becoming infected and actively passing the virus to older animals and then to the environment. Reorganization and reshaping of the camel industry includes allocating the camel markets away from the cities. Global awareness concerning the necessity of thorough boiling and cooking of the camel milk and meat products, respectively, should take place. Animal abattoirs should be established far away from large cities. They should not use mixed-animal platforms, and each platform should deal with one species of animal. Thorough decontamination of animals' biological wastes in abattoirs should occur using the appropriate standard protocols. Regular surveillance of MERS-CoV among the population especially during the active peak of virus shedding by the animals should occur during November to April every year. People who are in close contact with the camels should wear proper personal protective equipment at all times.

## CONCLUSIONS

At almost 7 years after its emergence, there are ongoing reports of MERS-CoV infection from time to time. This may be related to many unknown aspects of the viral evolution and pathogenesis. Some of these unknown aspects are the following. (1) Little is still known about virus/host interactions and how MERS-CoV hijacks the host immune system. (2) Potential reservoirs in the context of the MERS-CoV/human/animal network need to be identified. (3) Why do MERS-CoV-infected dromedary camels not exhibit obvious clinical signs during active viral shedding? (4) Does the presence of neutralizing antibodies in the sera of animals protect them against future active infection by the virus? (5) Does

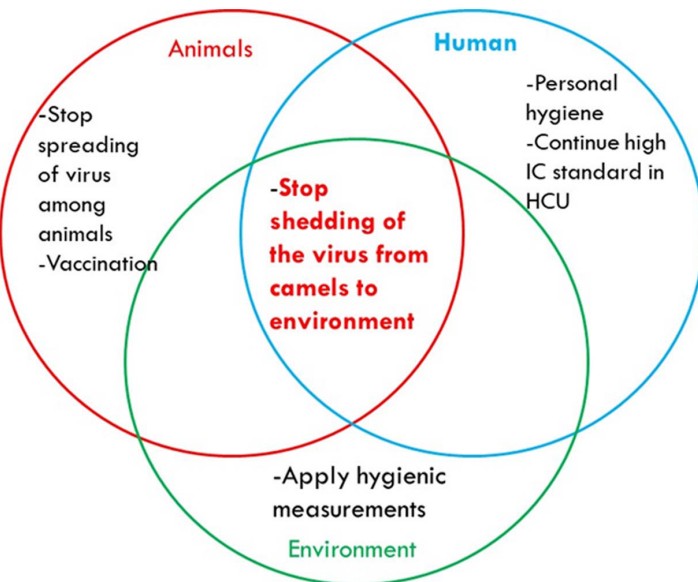

**Figure 2** **Some interventions based on the One Health-based to stop MERS-CoV outbreaks.**

vaccination of dromedary camels have a positive impact on controlling the spread of MERS-CoV among dromedary camels? More research is urgently needed to explore the unknown aspects of the MERS-CoV/human/animal network.

### Funding
This work was funded by a grant from the King Abdul-Aziz City for Science and Technology (KACST), through the MERS-CoV research grant program (number 20-0004), which is part of the Targeted Research Program (TRP). The funders had no role in study design, data collection and analysis, decision to publish, or preparation of the manuscript.

### Grant Disclosures
The following grant information was disclosed by the author:
King Abdul-Aziz City for Science and Technology (KACST).
MERS-CoV research grant program: 20-0004.
Targeted Research Program (TRP).

### Competing Interests
The author declares there are no competing interests.

### Author Contributions
- Maged Gomaa Hemida conceived and designed the experiments, performed the experiments, analyzed the data, contributed reagents/materials/analysis tools, prepared figures and/or tables, authored or reviewed drafts of the paper, approved the final draft.

## Data Availability

This is a literature review; there is no raw data.

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
