# Peer review of "Middle East Respiratory Syndrome Coronavirus and the One Health concept"

_PeerJ, doi:10.7717/peerj.7556_

## Round 0.1 · original submission · Major Revisions

Please review and respond to the reviewer comments and suggestions.

Reviewer 1 ·

Basic reporting

This manuscript covered a relatively broad topic and aimed to discuss multiple aspects in Middle East respiratory syndrome coronavirus (MERS-CoV) field. The manuscript itself is quite extensive which deserves appraisal. However, this manuscript lacks focus rendering the messages to be unclear.

It is important to acknowledge that there are already several reviews out there on animal/human interaction aspect of MERS-CoV. One of this review is written by Dr. Hemida and colleagues, i.e. https://onlinelibrary.wiley.com/doi/full/10.1111/tbed.12401. Dr. Hemida’s current manuscript seems to offer a slightly different twist, i.e. using One Health perspective to assess the current MERS-CoV situation. However, the One Health aspect does not seem to be clearly outlined in the abstract or in the manuscript. Even in section 2, line 83-88, the One Health concept is narrowed into animal/human interaction, making it quite overlapping with his previously published review.

I suggest the author do a careful check-up for grammatical errors throughout this manuscript. These mistakes might seem minor but sometimes could give a different meaning. For example, in line 50, the author wrote, “… I used to do a literature search of the most up-to-date published articles…”. I believe the author intended to write, “…I did a literature search…” instead. I also suggest omitting using “we”, since this manuscript is written by one person. In the abstract, the author used “we” quite a lot, but in the manuscript, it changes into “I”.

It will be prudent for the author to consult a scientific writing consultant or to collaborate with another expert in the field, to increase both the readability and the reliability of this manuscript. Besides that, rearranging the content of this manuscript to accommodate a more One Health perspective, and adding some insights/suggestions on One Health-based intervention to stop MERS-CoV outbreaks would be beneficial to increase the value of this manuscript.

Experimental design

Several statements written in this manuscript are not well-supported by current references and/or not carefully phrased thus likely causing misinformation. Due to a limited amount of time I have to review this manuscript, I would only give several examples on this point. I highly suggest the author, either alone or with the help of collaborators or scientific writing consultant, to carefully re-evaluate this manuscript to filter out these mistakes.

Line 145-146: "The RNA may act as a potential source of infection since MERS-CoV is a positive-sense RNA virus."
There is no evidence that RNA on its own is enough to transmit the virus.

Line 160-163: "Only some members of the Camelidae family (dromedary camels, alpacas, and llamas) appear susceptible to MERS-CoV infection, which occurs naturally in dromedary camels and experimentally in alpacas and llamas (Corman et al. 2014; Vergara-Alert et al. 2017)."
This statement is not true since alpacas and llamas can be naturally infected as well like dromedary camels, as reported in https://wwwnc.cdc.gov/eid/article/22/6/15-2113_article and https://www.sciencedirect.com/science/article/pii/S2352771418300053?via%3Dihub.

Line 166-168: "Whether the absence of detectable MERS-CoV antibodies in Bactrian camel sera is due to the tested animals’ geographical location in Mongolia far from the Middle East and Africa remains unknown." It is not clear what is unknown then.

Line 169-172: "Another possibility is that this might be due to genetic factors that contribute to the resistance of Bactrians to MERS-CoV infections, which requires further investigation but suggests that genetic factors play roles in susceptibility to the viral infection." It is not clear what the sentence means.

Line 172-173: "Experimental MERS-CoV infection in both alpacas and llamas showed a similar pattern to that of dromedary camels (Vergara-Alert et al. 2017)." This study does not do experimental infection in alpacas, thus the referenced study is not correct.

Line 188-190: "Meanwhile, the clinical profiles and viral shedding curve during the course of the MERS-CoV infection in these bats was highly similar to that in dromedary camels (Munster et al. 2016)." This statement is not fully correct nor supported by the reference. The amount of infectious virus shedding in bats is still less compared to dromedary camels.

Line 191-193: "Interestingly, another study revealed that MERS-CoV receptors (dipeptidyl peptidase-4, DPP4) were expressed in the respiratory and digestive tracts of some insectivorous bats (Vergara-Alert et al. 2017)." Wrong reference. This study is on livestock animals, not bats.

Line 214-217: "The roles of dromedary camel milk, meat 215 products, and by-products in MERS-CoV transmission is debatable. Experimentally introducing MERS-CoV into raw milk revealed little difference between the viral stock in the milk and that maintained in Dulbecco’s modified Eagle’s medium (DMEM) (van Doremalen et al. 2014)." The author should acknowledge that this study is performed with high virus concentration. Secondly, in this study, https://www.eurosurveillance.org/content/10.2807/1560-7917.ES2014.19.23.20829, viral RNA is detected in a limited concentration in camel’s milk. Thus, at this moment, the majority of experts in MERS-CoV field agree that the risk of virus transmission through milk is really low, as also stated in FAO report.

Line 260-263: "Seroprevalence of MERS-CoV was reported in dromedary camels from different countries in Africa and Asia (Ali et al. 2017; Hemida et al. 2014); however, feral camels from Australia and the Canary islands were found to be seronegative (Crameri et al. 2015). The reasons behind this phenomenon are unclear." The author stated previously that geographical location could be one of the important factors. So saying that this phenomenon is unclear without further explanation is not correct.

Line 274-275: "The genetic susceptibility of some human populations, especially of blood relatives is unclear in the context of MERS-CoV infection." The sentence is unclear. I assume the author would like to state there are some variations in susceptibility among humans. But it is important to remember that such variation does not always do to genetic differences. Other factors, such as age and underlying comorbidities, have been proposed in many papers to cause this variation between individuals. Thus, the sentence in line 274-275 above is not only unclear but also not well-supported by current evidence.

Line 275-276: "Meanwhile, development of a risk scoring 276 system for human exposure to dromedary camels is urgently needed." It is not clear what scoring system that the author meant, and it is not clear why such a scoring system will pose as a solution for MERS-CoV outbreaks in the field.

Line 305-321: the whole section 9. It is not clear how the five progress in the control of MERS-CoV could cause the drop in the number of cases being reported, especially the third to fifth progress being mentioned. The therapeutic and control approaches, for example, are mostly not implemented in the field yet.

Validity of the findings

The manuscript itself does not have an Introduction section. The aims of this manuscript, however, are slightly outlined in the methodology section in line 54-57, i.e. "I identified some important gaps in the research dealing with MERS-CoV/human/environment in the context of the One Health concept. Meanwhile, I summarized the current acceptable theories on the emergence and evolution of MERS-CoV. Finally, I highlighted the progress made for the control of MERS-CoV."

For the conclusion, the author wrote "Nearly 6 years after its emergence, reports of MERS-CoV infection are ongoing. This may be related to the many unknown aspects of viral evolution and pathogenesis. More research is urgently needed to explore the unknown aspects of the MERS-CoV/human/animal network." This conclusion leaves a lot to be desired and does not seem to have a clear added value to the current MERS-CoV field.

I suggest the author provide an introduction section in this review to clearly outline the expected outcome generated from this review or why it is important to have this review written and published. The conclusion, ideally, would be the main insights generated from this review, based on the outline/goal described in the introduction section.

Reviewer 2 ·

Basic reporting

This review discusses the One Health concept as it relates to MERS-CoV. This topic was previously reviewed in 2017 by Dr. Haagman's group, who did a wonderful job summarizing different aspects of MERS-CoV circulation and different One Health intervention strategies to control transmission.

Experimental design

The author reports a methodology consistent with a comprehensive, unbiased coverage of the subject. However, this reviewer wonders if key manuscripts have been omitted from the review. For example, multiple group have looked at the potential role that goats, sheep, and horses could play but none of these references are cited. These species would be a critical part of the One Health concept and need to be addressed. Additionally, the author is missing citations describing screening in Bactrians on line 165 as well as a seropositive 'naturally infected' alpaca on line 163.

There are multiple sections that may be incorrectly cited. Please double check the citations on lines 145, 153, 1576.

This review is organized into multiple sections that flow nicely. However, each section needs to be broken down into multiple paragraphs. A few examples include:
- MERS-CoV in humans: This section starts with a description of MERS-CoV in humans, moves to viral genetics and circulation, and then back to clinical information. This section could be strengthened by only describing human clinical data and moving the clinical information described around line 119 to the beginning of the section
- MERS-CoV in animals: line 194 - experimental models should be moved to a different section and expanded or removed
- line 206, 232 can be broken into different paragraphs or removed

Validity of the findings

This review focuses on the One Health concept as it relates to MERS-CoV. This relationship is extremely important and should absolutely be explored.

However, the author makes a few claims that may not be supported by primary literature.
- The author mentions multiple times that the research rate has declined - this should be removed.
- line 151: the authors suggest establishing an infection model but then describe doing this in naturally infected populations – is this really an infection model?
- The author mentions multiple times that different exposure rates are likely due to genetic factors. To this reviewer's knowledge, there is no published research supporting this. Human differences could be explained by cultural, medical, etc factors. Additionally, a lack of positive samples in camels could be due to a lack of exposure - the review should be modified to reflect this. Some of these claims are purely speculative, and should either be modified with supporting evidence or labeled as speculation.
- The author writes about an undiscovered reservoir, but does not offer any evidence supporting this.
-

Additional comments

A lot of really good work clearly went into this review.

There are a few areas that could be improved with additional editing for English grammar.

For example:
- run-on sentences: examples on lines 62, 170
- the tense is not consistent: example on line 74
- grammar editing: examples on lines 12, 184, 186, 229, 281, 307-309
- line 203: MERS is the human syndrome, MERS-CoV is the virus
- line 142: please change 'runny nose' to an appropriate clinical description

---

## Round 0.2 · Major Revisions

We were unable to get both prior reviewers to re-review, but as this is an important topic I felt it was important to ask new reviewers to comment on your revision.

Please review the reviewers' suggestions. I look forward to your revised manuscript.

Reviewer 2 ·

Basic reporting

This manuscript's grammar and flow has been significantly improved. However, I believe more editing will further improve this work. A few examples:

- Capitalization should be standardized (Middle East Respiratory Syndrome Coronavirus vs Middle East respiratory syndrome coronavirus)

- Line 78: "Two of them" should be reworded

- Line 105: "the" should be removed in front of MERS-CoV and SARS-CoV

- Line 148: Citations should be consistent throughout and either numbered or named. Italics throughout should also be corrected for consistency.

Line 277: The author should either list the brucellosis species of interest or change Brucella to brucellosis

This manuscript still needs to be further organized into smaller paragraphs so it is easier to follow. For example, Section 4 contains one paragraph that is over 25 sentences long and is nearly 2 pages.

Experimental design

The survey methodology and sources cited have been significantly improved. However, there are several areas that can still be improved.

Line 98: The author should remove the sentence about a sharp decline in research unless there is data supporting it. New manuscripts are frequently being released and I’m not sure that this is a fair characterization.

Line 103: One Health was first described in 400 BC and is not a new concept

Some of the data cited is still incorrect. The author should return to the original manuscripts to verify they are cited appropriately.

Validity of the findings

The author speculates about a missing animal reservoir; however, does not include any data to support this claim and has omitted several manuscripts exploring this topic through field sampling or experimental infections.

Additional comments

This manuscript has been significantly improved; however, I feel that more improvements are required. The addition of text expanding on One Health and MERS-CoV were valuable, and I think has dramatically improved the manuscript.

Reviewer 3 ·

Basic reporting

This manuscript review the historical background related to MERS-COV and place some target priorities for future research on this important viral diseases. The review discuss possible routes of transmission and factors affecting the virus transmission and management.

Experimental design

Study design was poorly developed and require further details; the author identified PubMed Central, Google Scholar, and Researchgate as the literature repositories; however, they never refer to the search keywords or search algorithm implemented. Some overlaps may occur between these literature databases and author should make clear how they avoided similar overlap in literature and how much unique literature he obtained from each database. The author should show a flow diagram of literature search for systematic review. Finally, i suggest author to consider using PRISMA checklist and flowchart.

Validity of the findings

The review require much details; however, it is successfully identified some research-related gaps of MERS-COV.

Additional comments

Line 30: Authors stated "other unknown reservoirs". I think that only a very poor evidence is available for bats as reservoirs (i.e. very short sequence similarity); however, MERS-Cov was reported from camels both in vivo and in vitro. Then, what other reservoirs are possibly involved?

Line 35: Replace "ex vivo" to "in vivo".

Line 42-43: Author should provide more details to support this conclusion. I would suggest author to add a reference to these information.

Line 53-59: The introduction section is poorly developed and should provide further details on MERS-Cov. Introduction presented only objectives of the review and none of historical background was presented as stated by the authors on Line 63-64.

Line 64: Author should introduce the "One Health concept".

Line 79-80: "This is a relatively short period for the emergence of a new coronavirus". How authors draw this conclusion?

Line 87: Remove ", especially from the Middle East".

Line 91-92: Cite a reference to the statement "There are many uncertain aspects of the virus evolution, pathogenesis, and transmission cycle".

Line 93-94: I don't agree with this statement. I can see a lot of progress on MERS-cov research and the virus attracted attention by FAO. I think one of the main critique to this review is the absence of any of the available FAO reports.

Line 98-102: I think author can provide more details in this context; the information mentioned herein was not sufficient to link both MERS-cov and the "One Health concept". I would suggest the authors to go more deep to the role of animals as a reservoir, particularly in case of MERS-Cov.

Line 112: Do you really mean ex vivo tissue culture or "in vivo".

Line 129: I think these evidences are limited only to positive antibody samples, so, the virus antibodies only were reported but none of the virus RNA.

Line 156: Correct "it being recognized".

Line 167: Replace "This" with "these".

Line 203: Cite a reference here for this study.

Line 216-219: The presence of antibodies doesn't necessarily reflect the role of these animals as reservoirs; however, it reflects only historical exposures.

I think author should show a flowchart for the future research priorities.

Line 331-343: Should you consider the paper published by Reeves et al. 2015. I think that the authors of this publication touched some important points related to MERS-COV geography and epidemiology.

Reviewer 4 ·

Basic reporting

no comment

Experimental design

No comment

Validity of the findings

No comment

Additional comments

I’m really appreciate the great effort of the author in both selection of topics and text. The author try to complete the story of MERS-CoV, He started with immunological aspects and MERS-CoV vaccine in his previous article and here the author tried to summarize all items related to MERS-CoV (12 subtitle) in this report. This extensive summary made the subtitles are much wide than their text and overlapping was common in the text. The lack of focusing make the message unclear.
It is well known now that the main issue in MERS-CoV is the transmission mechanism and origin. I recommend that the author focus his review on these topics, it well be more useful.
I see that the one health concept was broken into the title. The one health context is absent in the report according to CDC definition ( https://www.cdc.gov/onehealth/ and One Health Initiative http://www.onehealthinitiative.com/about.php.)
The author avoid to deals with the genetic characteristics of MERS-CoV isolated in East Africa and The Arabian Peninsula which will be the key to understand the future outbreaks.
The expected evolution of the virus and its impact on humans must be involved in the report.

L27….. you wrote “MERS-CoV is considered an ideal example of the One Health concept” How?
L 88…. The total No. of MERS-CoV human cases needs to be updated to 2434.
L 233…. Transmission of MERS-CoV from dromedary camels to humans has been proven indirectly in
some recent reports (Azhar et al., 2014). I think that 2014 not recent, please remove “recent”

---

## Round 0.3 · accepted · Accept

Thank you for working through the peer-review process. I believe your paper is much improved.

Reviewer 3 ·

Basic reporting

No comment

Experimental design

No comment

Validity of the findings

No comment

Additional comments

All comments have been addressed properly.

Reviewer 4 ·

Basic reporting

The introduction is focused on the aims of writing this review and the text is collective to the knowledge needed for understanding of MERS-CoV dynamic. The structure of the article is good and simple.

Experimental design

The study within the aims and scope of the journal and the cited reference are adequate and sufficient.

Validity of the findings

The article revise many of the knowledge needed for MERS-CoV and its circulation in the Middle East and world.